# Ex vivo pretreatment of human vessels with siRNA nanoparticles provides protein silencing in endothelial cells

Jiajia Cui[1], Lingfeng Qin[2], Junwei Zhang[3], Parwiz Abrahimi[4], Hong Li[1], Guangxin Li[2], Gregory T. Tietjen[1], George Tellides[2], Jordan S. Pober[4] & W. Mark Saltzman[1,3]

Human endothelial cells are initiators and targets of the rejection response. Pre-operative modification of endothelial cells by small interfering RNA transfection could shape the nature of the host response post-transplantation. Ablation of endothelial cell class II major histocompatibility complex molecules by small interfering RNA targeting of class II transactivator can reduce the capacity of human endothelial cells to recruit and activate alloreactive T cells. Here, we report the development of small interfering RNA-releasing poly (amine-co-ester) nanoparticles, distinguished by their high content of a hydrophobic lactone. We show that a single transfection of small interfering RNA targeting class II transactivator attenuates major histocompatibility complex class II expression on endothelial cells for at least 4 to 6 weeks after transplantation into immunodeficient mouse hosts. Furthermore, silencing of major histocompatibility complex class II reduces allogeneic T-cell responses in vitro and in vivo. These data suggest that poly(amine-co-ester) nanoparticles, potentially administered during ex vivo normothermic machine perfusion of human organs, could be used to modify endothelial cells with a sustained effect after transplantation.

[1] Department of Biomedical Engineering, Yale University, New Haven, CT 06511, USA. [2] Department of Surgery, Yale University School of Medicine, New Haven, CT 06511, USA. [3] Department of Chemical Engineering, Yale University, New Haven, CT 06511, USA. [4] Department of Immunobiology, Yale University School of Medicine, New Haven, CT 06511, USA. Correspondence and requests for materials should be addressed to W.M.S. (email: mark.saltzman@yale.edu)

A pproximately 25,000 organ transplants are performed each year in the United States, and 130,000 more patients are on the waitlist for an organ[1]. For patients diagnosed with end-stage kidney, liver, heart, or lung failure, organ transplantation is the only definitive long-term treatment option. Allografts are still subject to acute and chronic rejection, demonstrated by reduction in graft survival over time[2, 3]. Immunosuppressive therapy reduces the risk of rejection in the peri-transplant period where rejection is at the highest risk of occurrence; however, this approach is associated with major adverse effects such as infections, malignancies, bone marrow suppression, and cardiovascular toxicities[4, 5]. An alternative approach is to modify the graft perioperatively to reduce its capacity to activate the immune system during this period.

Human endothelial cells play a critical role in transplant rejection. Graft endothelial cells can initiate graft rejection by presentation of immunomodulatory proteins, such as class I and class II major histocompatibility complex (MHC) alloantigens, costimulators, and cytokines, to circulating host effector memory T cells[6–8]. Modifying graft endothelial cells to reduce MHC molecule expression can complement the anti-rejection benefits of both standard induction therapy, which provides a period of severe immunosuppression in the peri-transplant period, and removal of preformed donor-specific antibody, without further compromising the host's immune system[9].

The key problem faced in applying this approach to clinical practice is how to safely and effectively reduce MHC molecule expression on graft endothelial cells at the time of transplantation. Small interfering RNA (siRNA) can transiently reduce protein expression in the allograft[10]. Since acute rejection episodes are a risk factor for chronic rejection and late graft loss, reduction of rejection in the peri-operative period could reduce the risk of chronic rejection as well[11]. However, delivery of siRNA to endothelial cells is complicated by poor stability and limited membrane permeation of RNA[12–14]. Many prior attempts have been made to engineer delivery systems for siRNA, often by using cationic polymers or lipids that form nano-scale complexes with negatively charged nucleic acid[12–16]; these approaches are effective in vitro, but they exhibit significant cytotoxicity. Moreover, the duration of gene silencing is usually limited to 2–3 days[12, 13, 15, 16], which is insufficient for peri-operative inflammation to resolve. Polymer nanoparticles, such as poly (lactide-co-glycolide) (PLGA), are not toxic, and they can be loaded with substantial quantities of siRNA[17], but these materials have low encapsulation efficiency and limited transfection efficiency[14, 18]. Recent work using lipid-polymer hybrid nanoparticle-mediated transection of siRNA into human endothelial cells has been limited to in vitro studies[19, 20]. Here, we describe a biodegradable poly(amine-co-ester) (PACE) nanoparticle that demonstrates high encapsulation efficiency (~75%) and long-lasting protein knockdown in human endothelial cells both in vitro and in vivo without causing toxic effects in the transfected cells. Our laboratories recently reported that ablation of endothelial cell MHC class II molecule expression can prevent CD4 + effector memory T-cell activation, depriving CD8 + effector memory cells of help required to differentiate into cytotoxic T lymphocytes (CTLs), thereby protecting endothelial cells from CTL-mediated destruction in vivo[10]. Delivery of siRNA that targets the expression of class II transactivator (CIITA), a positive regulator for the transcription of MHC class II molecules, produces a brief period of refractoriness to interferon (IFN)-γ-mediated induction of MHC class II molecules. The present study was designed to test the feasibility of using siRNA-loaded PACE nanoparticles to silence immunomodulatory proteins on graft endothelial cells to reduce their capacity to activate the immune system for a sustained period of weeks, comparable to that

**Fig. 1** Synthesis of PACE polymer. PACE was synthesized through enzyme catalyzed copolymerization of 15-pentadecanolide (PDL), diethyl sebacate (DES), and N-methyldiethanolamine (MDEA). This polymerization reaction was carried out at 90 °C and at 1 atm of argon gas. After 24 h, pressure was reduced to 1.6 mm Hg and reaction continued for another 72 h. Polymer was then purified in hexane to remove the residual solvent and the dissolved in methylene chloride and filtered to remove residual catalyst

achieved by induction therapy or by antibody removal. We have again targeted CIITA as proof or principle, but we recognize that multiple molecules may need to be simultaneously targeted to get the full benefits of graft modulation.

Pre-transplant perfusion presents an unique opportunity to deliver siRNA-loaded nanoparticles to the allograft endothelium ex vivo[21]. Ex vivo normothermic machine perfusion (NMP) is a recently developed method of improving organ function prior to transplantation[22]. For many organs (kidneys, pancreas, and lungs), NMP has been used successfully to both preserve and re-condition organs for transplantation[22–24]. Here, we simulate NMP by perfusion through a single blood vessel with CIITA siRNA-loaded PACE nanoparticles and show effective and reliable particle uptake and MHC class II molecule silencing in the vascular endothelial cells. Furthermore, as proof-of-principle, we demonstrate that human artery segments pretreated ex vivo using siRNA-loaded nanoparticles are refractory to MHC class II molecule induction on endothelial cells for at least 4 to 6 weeks after transplantation into immunodeficient mouse hosts. Even with a single molecular target, this approach provides significant protection of human arterial grafts from rejection by adoptively transferred allogeneic T cells.

## Results

**Synthesis and characterization of PACE nanoparticles**. We synthesized biodegradable PACE polymers through enzymatic copolymerization of 15-pentadecanolide (PDL), diethyl sebacate (DES), and N-methyldiethanolamine (MDEA) (Fig. 1)[25]. PDL composition was varied based on moles of PDL fed to the reactor divided by the moles of PDL plus DES fed. PDL composition ranged from 50 to 90% (PDL %) and PACE polymers with molecular weights greater than 30 k were obtained (Table 1). Nitrogen content in the polymers decreased from 2.7 to 0.6%, as the PDL content increased from 50 to 90%. The PDL composition (%) in each PACE polymer was used to identify it: i.e., PACE-50 is PACE with ~50% PDL.

In previous work, we used PACE with a low PDL content (10 or 20%) to form polyplexes with plasmid DNA[25]: these polyplexes had low toxicity and high transfection efficiencies in cultured cells and in animals. We hypothesized that PACE with

### Table 1 Characterization of PACE polymers

| Name | PDL:DES:MDEA (feed molar ratio) | PDL:DES:MDEA (molar ratio, NMR) | $M_W$ | $M_W/M_n$ | Nitrogen content (wt%) |
|------|------|------|------|------|------|
| PACE-50 | 50:50:50 | 54:46:46 | 37,586 | 2.5 | 2.7 |
| PACE-60 | 60:40:40 | 62:38:38 | 46,811 | 3.3 | 2.2 |
| PACE-70 | 70:30:30 | 72:28:28 | 31,569 | 1.9 | 1.6 |
| PACE-80 | 80:20:20 | 82:18:18 | 35,400 | 2.2 | 1.0 |
| PACE-90 | 90:10:10 | 91:9:9 | 41,500 | 2.1 | 0.6 |

Polymers were synthesized using 50 to 90% PDL. % PDL was calculated based on PDL feed molar ratio (PDL:PDL + DES). PDL:DES:MDEA molar ratios in the resulting polymers were measured using NMR and polymer molecular weight was measured using GPC

### Table 2 Characterization of PACE NPs

| Name | Diameter (nm) | siRNA loading (pmol/mg) | Encapsulation efficiency (%) | Zeta potential (mV) |
|------|------|------|------|------|
| PACE-50 | 290 ± 73 | 358 ± 9 | 71.8 | 18.6 ± 0.3 |
| PACE-60 | 249 ± 61 | 456 ± 46 | 91.2 | 18.2 ± 0.1 |
| PACE-70 | 288 ± 58 | 350 ± 12 | 70 | 16.8 ± 0.1 |
| PACE-80 | 320 ± 73 | 461 ± 28 | 92.2 | 13.1 ± 0.7 |
| PACE-90 | 411 ± 82 | 383 ± 18 | 76.6 | 1.6 ± 0.02 |

CIITA siRNA-loaded NPs were produced from PACE polymers containing 50, 60, 70, 80, or 90% PDL using a modified double emulsion solvent evaporation technique. NP diameter was measured using SEM and NP surface charge was measured using Zetasizer (Malvern). siRNA loading in PACE NPs was quantified using a PicoGreen Assay and the encapsulation efficiency was calculated from the following equation: actual siRNA loaded/theoretical siRNA loaded × 100%

higher PDL content would be more hydrophobic and could be fashioned into solid nanoparticles using emulsion methods we developed to make PLGA nanoparticles[17, 26, 27]. To test this hypothesis, we used PACE-50, 60, 70, 80, and 90 to fabricate solid, biodegradable siRNA-loaded nanoparticles. PACE-50, 60, and 70 nanoparticles were smaller (diameter = 250–290 nm) than PACE-80 and 90 nanoparticles (diameter = 320–410 nm) (Table 2). Nanoparticles formed from PACE with lower PDL content (e.g., PACE-50), and resulting higher nitrogen content, had positive surface charges (zeta potential = 18 mV), while nanoparticles formulated from PACE with higher PDL content (e.g., PACE-90) had near-neutral surface charge (zeta potential = 1 mV) (Table 2). Similar quantities of siRNA (350–461 pmol mg$^{-1}$) were loaded into all PACE nanoparticles (Table 2). When the nanoparticles were incubated at 37 °C in buffered saline, the loaded siRNA was released over a period of 7 days in all formulations, with a burst release of ~80% of total siRNA over the first 2 days, followed by a gradual linear release of siRNA over the next 5 days (Fig. 2a).

**PACE nanoparticles demonstrate a range of cytotoxicity and transfection profiles in HUVECs.** We tested the siRNA-loaded PACE nanoparticles of different PDL composition for cytotoxicity and transfection of human umbilical vein endothelial cells (HUVECs) in vitro. Nanoparticles formed from PACE with the highest PDL content (PACE-90) exhibited negligible toxicity, even when added to cultures of HUVECs at high concentration (1 mg mL$^{-1}$ nanoparticles or 400 pmol mL$^{-1}$ siRNA) (Fig. 2b). Cytotoxicity was higher for other PACE-siRNA nanoparticles, but always less than an equivalent preparation of a commercially available transfection reagent, Lipofectamine (RNAiMAX): HUVECs treated with Lipofectamine demonstrated > 80% cell death at an siRNA concentration of 200 nM (Fig. 2b). The toxicity of the PACE nanoparticles correlated with PDL content and, therefore, the level of surface charge: PACE-90 nanoparticles were the least cytotoxic (~5% cell death at 1.5 mg mL$^{-1}$ of polymer, or 500 nM siRNA), PACE-70 nanoparticles were slightly more cytotoxic (30% cell death at 1.5 mg mL$^{-1}$ of polymer, or 500 nM siRNA), and PACE-50 nanoparticles were the most cytotoxic

(60% cell death at 1.5 mg mL$^{-1}$ of polymer, or 500 nM siRNA). Using fluorescent nanoparticles, loaded with DiD dye, and flow cytometry, we observed that PACE-50 nanoparticles (with the highest positive surface charge) associated strongly with HUVECs, whereas PACE-90 (with the near-neutral surface charge) had significantly lower association (Fig. 2c). These results indicate that cellular toxicity and association vary with PDL content of the PACE nanoparticles, presumably due to differences in surface charge on the nanoparticles, which is due to the density of primary amines on the polymer chain.

To examine whether PACE nanoparticles loaded with siRNA targeting CIITA can effectively suppress the expression of MHC class II proteins, we loaded siRNA directed against CIITA into PACE nanoparticles, and transfected cultured HUVEC. Nanoparticles made from PACE-50, 60, and 70 led to > 90% suppression of MHC class II protein expression, as measured by flow cytometry. In comparison, HUVECs treated with the same concentration of PACE-90 nanoparticles demonstrated 20% suppression of MHC class II expression (Fig. 2d). PACE 50, 60, and 70 nanoparticles were equally effective at silencing MHC class II expression (Fig. 2d), at least as effectively as lipofectamine, but PACE-70 nanoparticles demonstrated the lowest cytotoxicity of these three (Fig. 2b). Consequently, the rest of the studies in this report were performed using nanoparticles formed of PACE-70.

**PACE-70 nanoparticles suppress MHC class II expression in cultured HUVECs for 10 days.** When siRNA is delivered using conventional transfection reagents (i.e., Lipofectamine), the duration of protein knockdown is < 72 h[28]. Longer periods of protein suppression are likely to be needed to protect the transplanted allograft from recognition during the critical post-transplant period. To test for the duration of MHC class II suppression after nanoparticle administration, we examined the retention of siRNA-loaded nanoparticles in HUVECs. Nanoparticles were retained inside cultured cells for at least 11 days, although the number of nanoparticles in each cell appears to decrease with time (Fig. 3a). In cultured cells that are dividing, the number of nanoparticles present in each cell will be

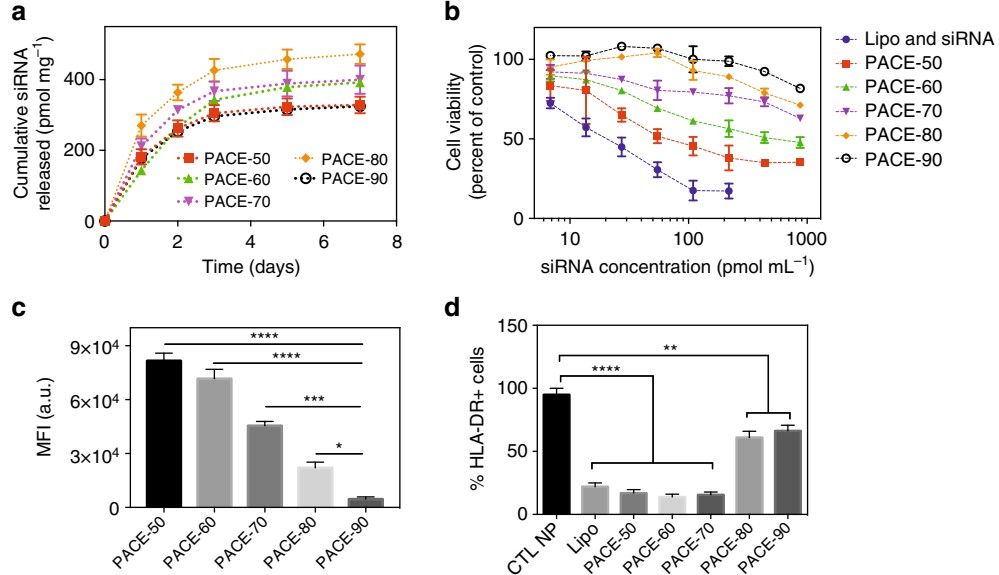

**Fig. 2** NP transfection and toxicity in HUVECs. **a** siRNA-loaded PACE-50, 60, 70, 80, 90 nanoparticles were incubated in PBS at 37 °C for up to 7 days. At each time point, PBS containing released siRNA was sampled and siRNA concentration was quantified using the PicoGreen Assay. **b** HUVECs were treated with siRNA PACE nanoparticles or lipofectamine (7–900 pmol mL$^{-1}$) for 24 h, after which cells were washed and cell viability was quantified using CellTiter-Blue Assay (Promega). **c** HUVECs were treated with DiD-loaded PACE nanoparticles for 6 h and nanoparticle association was quantified using flow cytometry. **d** Knockdown of HLA-DR (the highest expressed form of MHC II) protein on HUVECs was quantified using flow cytometry. HUVECs were treated with PACE nanoparticles (30 nM) or lipofectamine (30 nM). After 6 h, transfection agents were removed and cells were cultured in medium containing hIFN-γ for another 72 h. Data shown is mean ± SD ($n = 4$). **$p < 0.01$, ***$p < 0.001$, ****$p < 0.0001$ by one-way ANOVA

diluted naturally as a result of continuous cell division in vitro; for example, we know that fluorescent tracers such as PKH67 decrease in concentration exponentially during cell division (Fig. 3b). With nanoparticles and cultured cells, it is difficult to determine how much of the loss of nanoparticles observed by confocal microscopy is due to this natural dilution, as compared to other sources of loss. This is important, as we predict that the duration of protein knockdown provided by nanoparticles loaded with siRNA will depend on nanoparticle persistence in cells and sustained release (Fig. 2a). To see if the > 10 day persistence of nanoparticles in cultured HUVEC led to sustained knockdown, we measured the duration of MHC class II knockdown in HUVECs over time after a single treatment with nanoparticles. HUVECs treated with nanoparticles loaded with siRNA directed against CIITA showed a significantly longer period of protein knockdown compared to HUVECs treated using Lipofectamine (Fig. 3c): by day 5, > 70% of cells treated with Lipofectamine while < 10% of cells treated with nanoparticles had regained MHC class II molecule expression. Furthermore, MHC class II suppression continued for more than 10 days in HUVECs treated using PACE-70 nanoparticles. We observed no evidence of toxicity during this period.

**Ex vivo perfusion enhances nanoparticle uptake by vascular endothelial cells.** To test the applicability of these results to an intact tissue, we perfused nanoparticles – at flow rates ranging from 0 to 130 mL min$^{-1}$—through an isolated vessel (i.e., a human umbilical vein) for 6 h ex vivo. First, we confirmed that endothelial cells were retained in the tissue during perfusion under this range of flow rates (Fig. 4a). At perfusion rates up to 130 mL min$^{-1}$, the extent of endothelial cell coverage, calculated from CD31 + endothelial cell linear coverage post perfusion/CD31 + endothelial cell linear coverage pre perfusion, in the human umbilical vein remained intact during 6 h of perfusion (Fig. 4b). When veins were perfused with M199 culture medium containing siRNA-loaded nanoparticles, transfection—as

measured by the mean fluorescence intensity due to nanoparticles associated with HUVEC by flow cytometry—was optimal at an intermediate flow rate: we observed a fourfold increase in nanoparticle uptake at a flow rate of 16 mL min$^{-1}$ compared to 0 mL min$^{-1}$ and 130 mL min$^{-1}$ (Fig. 4c, d).

**Ex vivo nanoparticle pretreatment reduces CD4 + memory T-cell proliferation.** We further examined if ex vivo perfusion with siRNA-loaded nanoparticles can suppress MHC class II expression on endothelial cells in an intact tissue. To accomplish this, we perfused umbilical veins with nanoparticles, isolated and cultured the endothelial cells lining the vessel, and then exposed the cells to IFN-γ, which stimulates expression of CIITA (Fig. 5a). Endothelial cells harvested from veins perfused with siRNA-loaded nanoparticles demonstrated > 90% reduction of MHC class II expression compared to endothelial cells harvested from veins perfused with control nanoparticles (Fig. 5b–d). Additionally, treatment with siRNA-loaded nanoparticles specifically downregulated MHC class II expression, while preserving expression of other endothelial cell proteins, such as constitutively expressed CD31, IFN-γ-responsive MHC class I, E-selectin, and VCAM-1 (Supplementary Fig. 1). Next, we studied if suppression of MHC class II protein on endothelial cells in an intact vessel can attenuate CD4 + memory T-cell proliferation. CD4 + memory T cells cocultured with endothelial cells pretreated with siRNA-loaded nanoparticles demonstrated reduced proliferation (1.6%) compared to CD4 + memory T cells cocultured with endothelial cells pretreated with control siRNA-loaded nanoparticles (7.7%) (Fig. 5e).

**NPs suppress MHC class II expression for up to 6 weeks in transplanted human arteries in vivo.** To test whether ex vivo treatment of tissue would lead to long-lasting knockdown of CIITA after transplantation, we pretreated human arterial allografts with siRNA-loaded nanoparticles for 6 h ex vivo before transplanting these arteries into immunodeficient SCID/beige

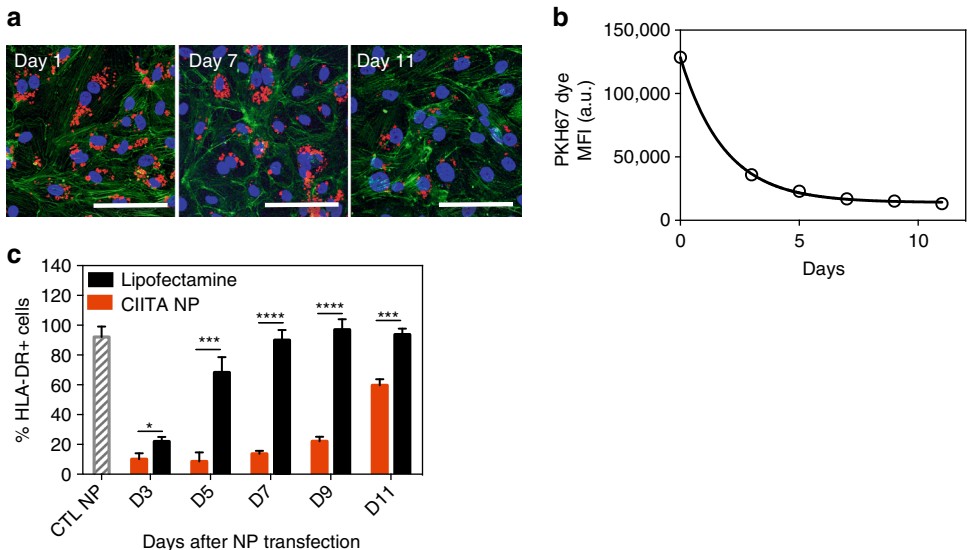

**Fig. 3** Long-term NP retention and MHC class II knockdown in HUVECs. **a** siRNA-loaded nanoparticles (*red*) are retained in HUVECs for at least 11 days. Cells were stained using phalloidin (*green*) and dapi (*blue*) and imaged using a confocal microscope (*scale bar*: 100 μm). **b** Dilution of fluorescence over 11 days in HUVECs pre-labeled with PKH67 tracer (Sigma). Labeled HUVECs were seeded in a 24-well plate at a concentration of $10^5$ cells per well. Cell tracer fluorescence was measured using a flow cytometer. **c** Durable knockdown of HLA-DR protein on HUVECs for 11 days. HUVECs were treated with siRNA-loaded nanoparticles or Lipofectamine for 6 h and cultured for up to 11 days. At days 3, 5, 7, 9, and 11, cells were harvested and surface HLA-DR expression was quantified using a flow cytometer. Data shown is mean ± SD ($n = 4$). **$p < 0.01$, ***$p < 0.001$, ****$p < 0.0001$ by Student's *t*-test

mouse hosts. One week after transplant, nanoparticles were still associated with endothelial cells in the human arterial allograft and could not be detected in other mouse organs (i.e., liver, spleen, lungs, aorta) (Supplementary Fig. 2). We next investigated if pretreatment with siRNA-loaded nanoparticles ex vivo is sufficient to knockdown MHC class II expression in human arterial allografts for up to 6 weeks after transplantation (Fig. 6a). Remarkably, human arterial allografts pretreated with siRNA-loaded nanoparticles showed > 80% suppression of MHC class II expression at 1 week and 2 weeks, > 45% suppression of MHC class II expression at 4 weeks, and ~20% suppression of MHC class II expression at 6 weeks after transplant (Fig. 6b, c).

**MHC class II knockdown reduces intimal expansion and T-cell infiltration**. To evaluate whether knockdown of MHC class II on human arterial endothelial cells would suppress T cell- mediated allograft destruction in vivo, we transplanted nanoparticle-pretreated human arterial segments into mice previously inoculated with human T cells allogeneic to the artery donor and evaluated arterial graft injury 10 days later (Fig. 7a). Specifically, 1 mM diameter adjacent segments of human epicardial coronary artery were engrafted into the infrarenal aortae of immunodeficient C.B-17 SCID/bg mice (one segment per mouse host), allowed to heal in for 30 days, at which point the artery segments were harvested, treated with siRNA-loaded nanoparticles ex vivo, and each re-transplanted into a second SCID/bg mouse that had been previously inoculated intraperitoneally with human peripheral blood mononuclear cells (PBMCs) allogeneic to the artery donor[10, 29, 30]. In this model, the adoptive transfer of human PBMCs results in the engraftment of circulating human CD4 + and CD8 + T cells, that by 2 to 3 weeks constitute 5–10% of the total blood mononuclear cell population[30]. By 10 days post-artery transplantation, the circulating human T cells infiltrate the graft and induce intimal expansion and injury. Knockdown of MHC class II expression on human arterial allografts treated with CIITA siRNA nanoparticles resulted in significantly reduced intimal area and vessel wall thickness compared to control siRNA nanoparticle treated arteries (Fig. 7b).

Arteries treated with CIITA nanoparticles also retained significantly greater CD31 + endothelial cell coverage and mural cell α-smooth muscle actin (SMA) expression compared to arteries treated with control nanoparticles (Fig. 7c). Additionally, arteries treated with CIITA nanoparticles demonstrated reduced intimal T-cell infiltration (Fig. 7d). Altogether, our findings suggest that MHC class II knockdown confers a protective effect to both allograft endothelial and smooth muscle cells although only the endothelial cells actually express MHC class II molecules. The protective effect presumably occurs because class II-deficient endothelial cells are unable to activate allogeneic CD4 + T cells, depriving the CD8 + T cells of "help" required for differentiation into cytotoxic T cells, which mediate graft cell destruction.

## Discussion

Current anti-rejection medications, such as antibodies and small molecule drugs, target the immune systems of the organ recipient, which can cause substantial systemic effects. Therapies targeted to the organ prior to transplantation remain unexplored and might be safer. Modulation of the graft to minimize rejection may allow reduction in the level of systemic immunosuppression required to prevent rejection. Here, we present a method for sustained knockdown of proteins in the endothelial cell of an intact tissue. When we used this approach to silence a protein critical for immune recognition on the surface of endothelial cells (MHC class II), CD4 + memory T-cell activation on exposure to these cells was markedly attenuated. Since it has been previously demonstrated that ablation of MHC class II on endothelial cells can reduce CD4 + memory T cell-help required for human CD8 + CTL differentiation and graft destruction[10], we believe that this approach, when used in conjunction with systemic therapies, will be useful in protecting the transplanted organ from rejection. In these proof-of-principle studies, we chose to test the efficacy of PACE nanoparticle-mediated transfection with siRNA that targets CIITA because of prior studies in which we anticipated a beneficial effect. The full benefit in clinical transplantation may involve simultaneously silencing of multiple molecules, including MHC class I molecules that are the target of CD8 + cytotoxic

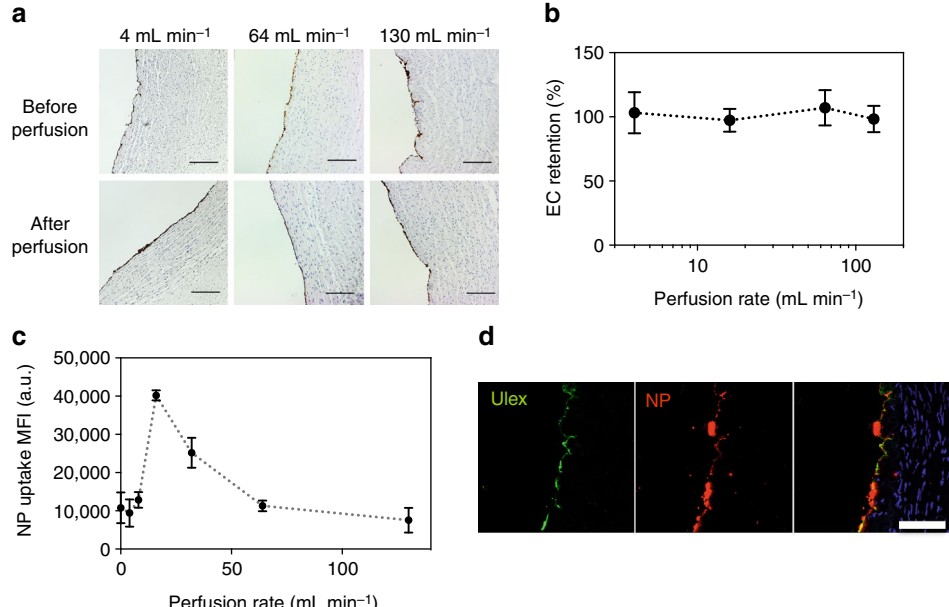

**Fig. 4** Perfusion enhances NP uptake by vascular endothelial cells. **a** Endothelial cell are retained on umbilical veins after perfusion. Veins were perfused with culture media for 6 h using a peristaltic pump (Langer Instruments). After perfusion, vein sections were fixed in 4% PFA, sectioned, and stained for human CD31 (*scale bar*: 200 μm). **b** Retention of endothelial cells (ECs), expressed as a %, was calculated from: % EC coverage after perfusion/% of EC coverage before perfusion. **c** NP (*fluorescent*) association with umbilical veins after perfusion. Veins were perfused with DiD-NPs (0.1 mg mL$^{-1}$) at 0–130 mL min$^{-1}$. After 6 h, ECs were harvested from the vein and particle-cell association was quantified using flow cytometry. Data shown is mean ± SD. **d** Confocal image of *fluorescent* NPs (*red*) associated with endothelial cells (*green*) on the umbilical vein after perfusion (16 mL min$^{-1}$, 6 h), *scale bar*: 100 μm

T cells and many preformed anti-donor antibodies. Our approach of targeting the graft is complementary to current approaches that target the host's innate and adaptive immune systems.

We have previously shown that PACE polymers, which have low cation densities, can produce polyplexes with plasmid DNA that provide efficient transfection of cells, both cultured in a dish and tumor cells growing in animals, without toxicity. To optimize and sustain siRNA delivery to endothelial cells, we extended our synthesis methods to produce a new family of PACE polymers. In our previous work, PACE synthesized with 10% PDL condensed plasmid DNA into nanosized polyplexes and transfected cells with improved efficiency[25]. Here, we increased the composition of PDL to 50–90%, which led to more hydrophobic polymers that could form solid core nanoparticles. When nanoparticles are formed in the presence of siRNA, PACE condenses and stabilizes siRNA through electrostatic interactions (via interactions with amines on the MDEA monomers) and hydrophobic interactions (via hydrophobic interactions among the lactone groups). Cationic amines (MDEA) within PACE appear to enhance the transfection efficiency of nanoparticles when they are exposed to cells. The presence of cationic amines may improve transfection via the following mechanisms: (1) cationic nanoparticles may associate more effectively with negatively-charged lipid membranes to facilitate uptake, (2) cationic nanoparticles may create osmotic swelling and lysis of endosomal compartments to facilitate escape into the cytoplasm via the proton-sponge effect[31–34].

In cultured cells that are actively dividing, MHC class II silencing persists for more than 10 days in vitro. In comparison, Lipofectamine suppressed only 70% of MHC class II expression for 3 days. When studied in dividing cells, the duration of MHC II suppression in vitro may be underestimated as a result of continuous cell proliferation (doubling time = 24 h) and dilution of intracellular nanoparticles and siRNA. Our attempts to prevent cell division in vitro for this length of time led to cellular toxicity. It is likely that, in the absence of cell proliferation, the duration of

protein suppression could be much longer. Fortunately, in the absence of injury, turnover of endothelial cells in situ is exceedingly low, and the effect of the administered and slowly released siRNA should last more than 10 days. Our findings confirms that PACE-70 nanoparticles continue to suppress IFN-γ-induced MHC class II molecule expression for up to 6 weeks in human arterial endothelial cells in vivo. Short-term suppression of endothelial cell immune stimulatory molecules during the perioperative period can have lasting effects on allograft outcomes because human effector memory T cells can be directed along different pathways by their initial contact with graft endothelial cells and the milieu in which this occurs.

Ex vivo normothermic perfusion (EVNP) provides an opportunity to deliver siRNA therapeutics to allograft organs prior to transplantation. Here, we developed a perfusion system to deliver siRNA-loaded nanoparticles to the human umbilical vein ex vivo. The rate of particle flow through a blood vessel plays an important role in the kinetics of nanoparticle—endothelial cell association: gentle flow (16 mL min$^{-1}$, shear stress = 0.1 dynes cm$^{-2}$) generates a fourfold increase in nanoparticle association compared to other tested flow rates; at higher flow rates, nanoparticle association decreases as a function of flow. A number of computational and in vitro studies demonstrated that nanoparticle association decreases as a function of increased shear stress (1–10 dynes cm$^{-2}$), which confirms our observation that the degree of nanoparticle—endothelial cell association decreases as shear stress increases (above 0.1 dynes cm$^{-2}$)[35–38]. Other studies, which examined the effect of sub–physiological shear stress (< 1 dynes cm$^{-2}$) on nanoparticle-endothelial cell association, reported that gentle flow (shear stress < 0.1 dynes cm$^{-2}$), compared to static incubation, reduces the sedimentation velocity of particles and increases frequency of particle collisions with endothelial cells, which explains our observation that nanoparticle association peaks at 16 mL min$^{-1}$[39]. In this proof-of-concept study, we illustrate that nanoparticle association can be optimized in an ex vivo human tissue perfusion system. Other

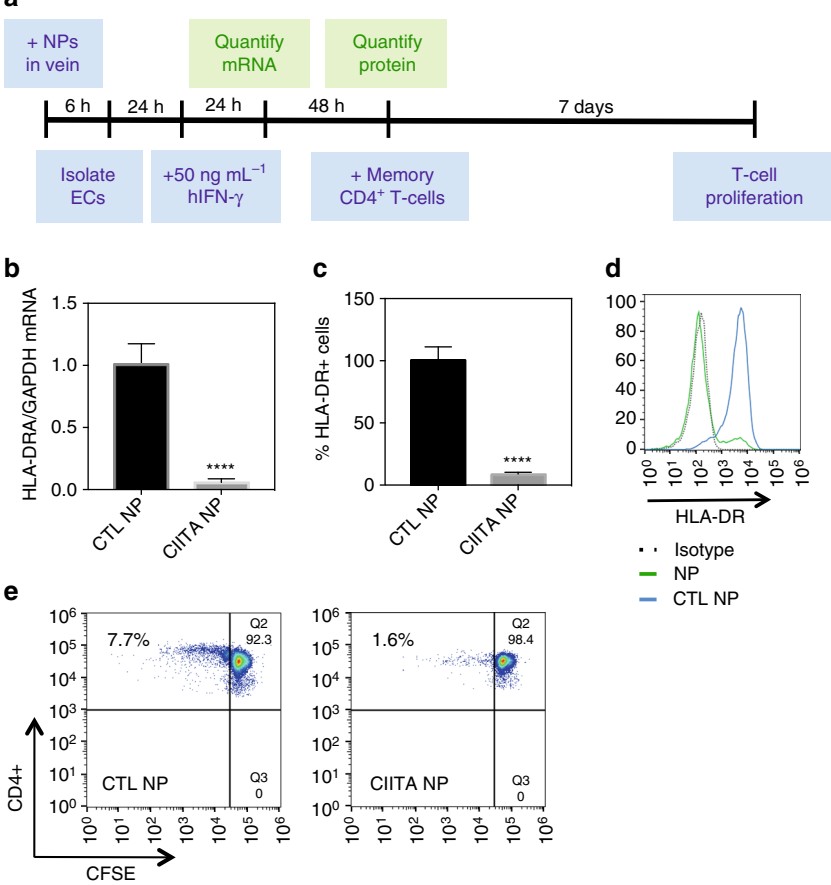

**Fig. 5** NP perfusion reduces CD4 + memory T-cell proliferation. **a** CD4 + memory T-cell proliferation. Umbilical veins were perfused with culture medium containing siRNA-loaded NPs (0.1 mg mL$^{-1}$) or CTL siRNA-loaded NPs (0.1 mg mL$^{-1}$). After 6 h, ECs were harvested from veins and seeded into 24-well plates. CFSE-labeled CD4 + memory T cells were co-cultured with endothelial cells for 7 days. T-cell proliferation was then quantified using flow cytometry **b** HLA-DRA/GAPDH mRNA was quantified 24 h after addition of hIFN-γ by quantitative reverse transcription-PCR. **c**, **d** HLA-DR protein was quantified by flow cytometry. **e** CD4 + memory T-cell proliferation was determined by CFSE dye dilution on flow cytometry. Data shown is mean ± SD ($n = 5$), ****$p < 0.0001$

parameters, such as particle size, surface charge, polymer composition, flow pressure, and duration, may also affect nanoparticle-endothelial cell association.

In conclusion, biodegradable PACE nanoparticles are a promising delivery platform for siRNA therapeutics because it provides effective encapsulation, transfection, controlled release, and low toxicity. We showed that short-term suppression of immunomodulatory proteins (i.e., MHC class II molecules) with siRNA-loaded nanoparticles on donor endothelial cells can protect the allograft from acute rejection after transplant. In the future, this technology can be used to target a combination of immunomodulatory proteins (i.e., MHC class I, LFA-3, raptor, cytokines) on donor endothelial cells to provide robust protection for the transplanted graft[29, 40].

## Methods

**Cell lines.** HUVECs were purchased from the Vascular Biology and Therapeutics Program (Yale University). HUVECs were isolated under a protocol approved by the Yale Human Investigations Committee. HUVECs were cultured in M199 media supplemented with 20% fetal bovine serum (FBS) (Invitrogen), 2 mM L-glutamine (Invitrogen), 100 U mL$^{-1}$ penicillin, 100 ug mL$^{-1}$ streptomycin, and 0.1% endothelial cell growth supplement (BD Bioscience, Franklin Lakes, NJ) on 0.1% gelatin coated tissue culture plates. In ex vivo perfusion experiments, HUVECs were isolated from umbilical veins by incubation in 0.1% collagenase IV (Worthington) for 15 min, and cultured on 0.1% gelatin coated tissue culture plates.

**Synthesis of poly(amine-co-ester).** PACE was synthesized through enzyme catalyzed copolymerization of 15-PDL, DES and MDEA, as described previously, with minor modifications to the composition of PDL (50–90%)[25]. In all, 15-PDL, DES, MDEA, Novozym 435 catalyst, and diphenyl ether solvent were stirred at 90 °C under 1 atm of argon gas for 24 h. The pressure was reduced to 1.6 mm Hg at 90 °C and the reaction was continued for another 72 h. The resulting polymer was purified in hexane to remove residual solvent, dissolved in methylene chloride, and filtered to remove the enzyme catalyst. The added methylene chloride was removed at 40 °C and 1.0 mm Hg. The filtrates were analyzed by GPC using polystyrene standards to measure polymer molecular weights.

**Fabrication of nanoparticles.** Nanoparticles were fabricated using a modified water-in-oil-in-water (w/o/w) double-emulsion solvent evaporation technique, as described previously[17, 18]. Briefly, 25 nmols of CIITA siRNA (5′-GAAGU-GAUCGGUGAGAGUAUU-3′) (GE Dharmacon) or nontargeting control siRNA (GE Dharmacon) was dissolved in pH 5.2 sodium acetate buffer containing 1 mM EDTA was added dropwise under vortex to 50 mg of PACE in methylene chloride, and sonicated to form the first water-in-oil emulsion. Next, the emulsion was added dropwise under vortex to a 5% PVA solution and sonicated to form the second water-in-oil-in-water emulsion. The particles were hardened in 0.3% PVA solution for 3 h, and washed in water three times to remove excess PVA. Particles were lyophilized for 48 h and stored in −20 °C prior to use.

**Characterization of nanoparticles.** Particle size and morphology were characterized using the scanning electron microscope (FEI, Hillsboro, Oregon) and quantified using ImageJ. Zeta potential was measured using the zetasizer (Malvern Instruments). siRNA loading was measured by dissolving nanoparticles in methylene chloride for 2 h, followed by siRNA extraction into pH 7.4 TE buffer (10 mM Tris-HCl, 1 mM EDTA) containing 5000 U mL$^{-1}$ heparin twice overnight.

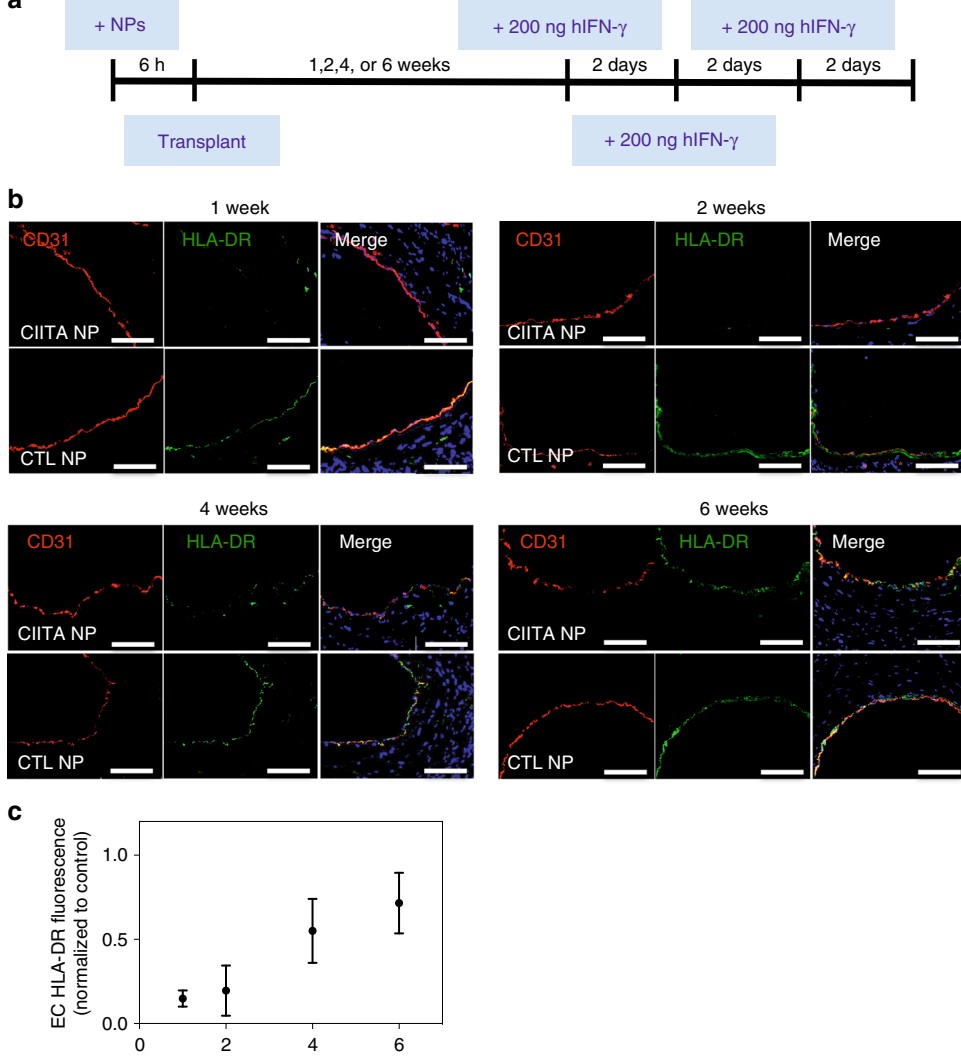

**Fig. 6** NPs suppress MHC class II expression for up to 6 weeks in transplanted human arteries in vivo. **a** Human arterial allografts pretreated with siRNA-NPs show reduction of HLA-DR on ECs for up to 6 weeks in vivo. Coronary arteries were incubated with siRNA-loaded NPs for 6 h and transplanted into SCID/beige mice. Three doses of hIFN-γ were administered every other day for 1 week. **b** After 1, 2, 4, or 6 weeks, arteries were harvested, sectioned, and stained using anti-CD3 antibody (*red*), anti HLA-DR antibody (*green*), and DAPI (*blue*), *scale bar*: 100 μm. **c** HLA-DR fluorescence was quantified from the images using ImageJ and normalized to CTL NP treated arteries. Data shown is mean ± SD (*n* = 4)

siRNA loading was quantified using QuantIT PicoGreen assay (Invitrogen) per manufacturer's instructions. siRNA released from particles was characterized by dissolving 1 mg of nanoparticles in PBS. The solution was incubated on a shaker at 37 °C and at each time point, samples were centrifuged for 5 min at 16,100×*g*. Supernatant containing released siRNA was collected and 1 mL of fresh PBS was replaced. siRNA concentration was quantified using QuantIT PicoGreen assay.

**Evaluation of nanoparticle cytotoxicity**. HUVECs were plated in 96-well tissue culture plates and incubated in M199 medium containing PACE-50, 60, 70, 80, 90 nanoparticles. After 24 h, nanoparticles were removed and cells were washed three times and incubated in 100 μL of M199 containing 20 μL of cell titer blue (Promega) for 2 h. Fluorescence was measured using the plate reader at 560/590 nm.

**Evaluation of nanoparticle uptake in vitro**. HUVECs were plated in gelatin coated 24-well tissue culture plates with coverslips and treated with TAMRA-labeled siRNA nanoparticles at a concentration of 0.1 mg mL$^{-1}$. After 6 h, nanoparticles were removed and HUVECs were washed three times with PBS, fixed in 4% paraformaldehyde for 10 min, permeabilized in 0.1% Triton-X 100 (Sigma) in 2% BSA (Sigma) for 30 min, and incubated in Alexa Fluor 488 phalloidin (Life Technologies) for 20 min as per the manufacturer's instructions. Samples were mounted using VECTASHIELD with DAPI (vector labs), and imaged on Leica SP5 confocal microscope.

**Reverse transcription polymerase chain reaction**. HUVECs were incubated with siRNA nanoparticles for 6 h and treated with 30 ng mL$^{-1}$ recombinant human IFN-γ (Invitrogen) to re-induce endothelial cells expression of MHC class II. After 24 h, mRNA was isolated using RNeasy kit (Qiagen) as per the manufacturer's instructions. Reverse transcription was performed using QuantiTect reverse transcription kit (Qiagen). Transcript levels were quantified on CFX96 Rea-Time system using CFX Manager Software (Bio-Rad) using the following probes from Applied Biosystems: HLA-DRA (Hs00219578_m1), CIITA (Hs00172094_m1), and GAPDH (Hs002758991_g1). Relative expression was calculated using the $2^{-\Delta\Delta Ct}$ and gene expression levels were normalized to GAPDH.

**Flow cytometry**. Surface protein expression was quantified using flow cytometry. Briefly, endothelial cells were harvested, washed in 1% BSA in PBS, and incubated with an antibody against HLA-DR (clone L243, Biolegend) for 30 min on ice. After antibody incubation, endothelial cells were washed three times using 2% BSA in PBS and analyzed on a Attune NxT Flow Cytometer. Other endothelial cell antibodies used for flow cytometry include HLA-A,B,C (Biolegend) and CD31 (Biolegend). All results were analyzed using the FlowJo software (FlowJo LLC).

**Ex vivo perfusion**. Umbilical cords were obtained under a protocol approved by the Yale Human Investigations Committee. Umbilical cords were collected after Caesarean sections from Yale-New Haven Hospital, transported on ice, and

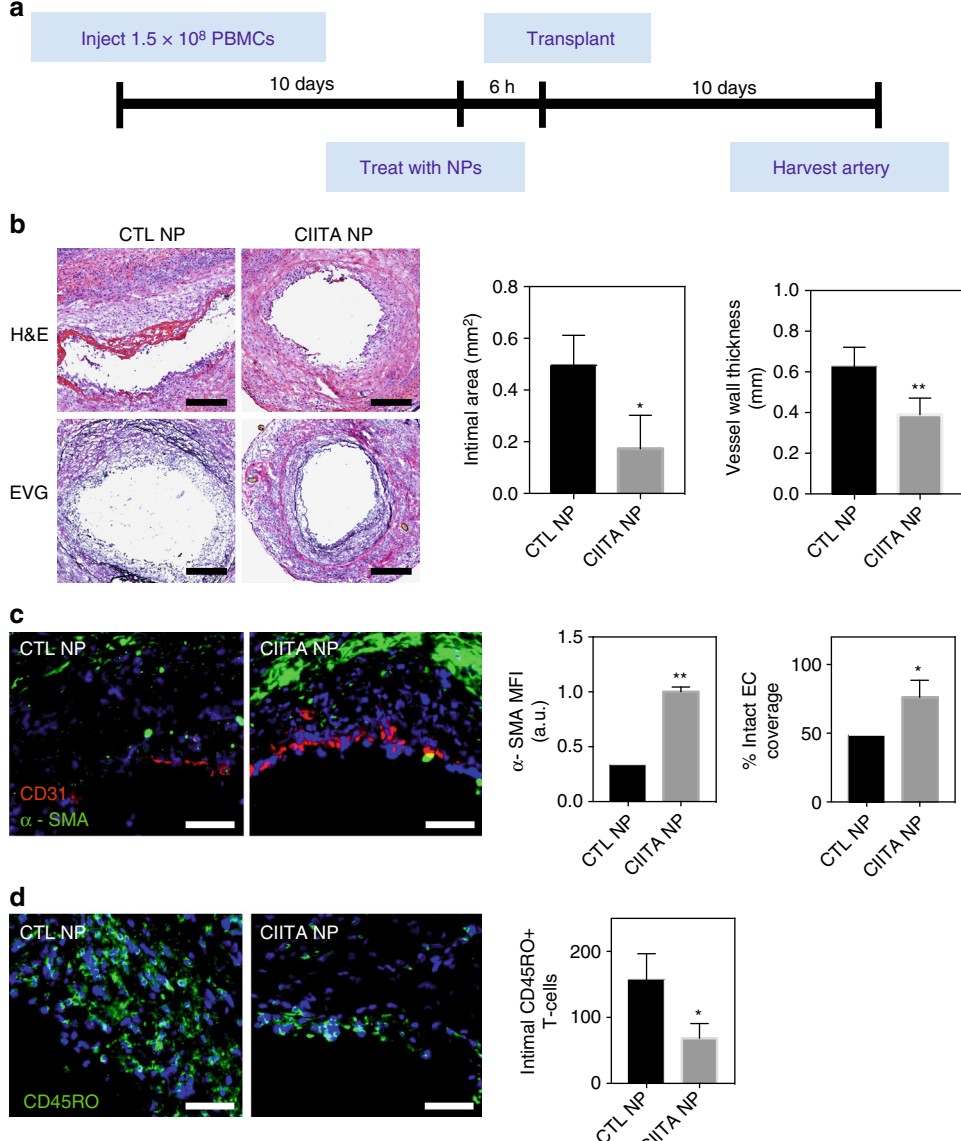

**Fig. 7** NPs reduce intimal expansion and T-cell infiltration in transplanted human arterial allograft. **a** Human arteries were pretreated with siRNA-loaded nanoparticles for 6 h and transplanted into SCID/bg mice previously reconstituted with $1.5 \times 10^8$ allogeneic human PBMCs. After 10 days, arteries were harvested, sectioned, and stained. **b** CIITA nanoparticle pretreated arteries ($n = 4$) or CTL nanoparticle pretreated arteries ($n = 4$) were sectioned and stained using H&E and EVG. Intimal area and vessel wall thickness were quantified using ImageJ (*scale bar*: 200 μm). **c** Artery sections were stained for human α-SMA (*green*), human CD31 (*red*), and DAPI (*blue*). α-SMA and endothelial cell coverage was quantified using ImageJ (*scale bar*: 100 μm). **d** Artery sections were stained for CD45RO (*green*) and DAPI (*blue*). Total number of CD45RO + cells in the intima (per cross-section) was quantified from fluorescent images (*scale bar*: 100 μm). Data shown is mean ± SD, *$p < 0.05$, **$p < 0.01$, by student's *t*-test

perfused within 2 h. Umbilical veins were connected to the perfusion system via barbed luers (Macmaster) and platinum-cured silicone tubing (Cole Parmer). Veins were submersed in M199 medium and kept at 37 °C during the experiment. Nanoparticles (0.1 mg mL$^{-1}$), resuspended in M199 medium, were perfused through the umbilical vein using a Low Flow Rate Peristaltic Pump (Langer Instruments). The flow rate was adjusted between 0 and 128 mL min$^{-1}$ (shear stress 0–0.8 dynes cm$^{-2}$) and the pressure was maintained at 5–10 mm Hg. Umbilical veins were perfused in this solution for 6 h. Percent of endothelial cell retention post perfusion was calculated as follows: CD31 + endothelial cell linear coverage post perfusion/ CD31 + endothelial cell linear coverage pre perfusion x 100%.

To collect cells for culture, flow cytometry, and /quantitative reverse transcription-PCR, HUVECs were isolated from umbilical veins[41]. In brief, after nanoparticle perfusion, umbilical veins were washed with HBSS x 3 and filled with 10 mL of 0.1% collagenase type II (Worthington). Veins were tied on both sides and incubated in a beaker of 200 mL Dulbecco's phosphate-buffered saline (DPBS) for 15 min at 37 °C. After incubation, veins were flushed with 20 mL of HBSS to collect dissociated endothelial cells. The cell solution was centrifuged at 250×*g* x 10

min to obtain a cell pellet. Isolated cells were stained for CD31 and gated for CD31 + cells on the flow cytometer, prior to quantification of fluorescent nanoparticle uptake.

**CD4 + memory T-cell proliferation assay**. PBMCs were collected by leukapheresis from anonymous volunteer donors and isolated using Ficoll Lymphocyte Separation Medium (MP Biomedicals). Purified PBMCs were cryopreserved in 10% DMS and 90% FBS in liquid nitrogen before use. CD4 + T cells were isolated from PBMCs using Dynabeads CD4 Positive Isolation Kit (Invitrogen) as per the manufacturer's instructions. To isolate memory T cells, naïve CD4 + T lymphocytes were removed by negative selection using anti-CD45RA (eBioscience) and depleted by incubation with Dynabeads Pan Mouse IgG (Invitrogen).

Isolated CD4 + memory T cells were stained with CellTrace CFSE dye (Invitrogen) at 0.5 μM for 15 min. HUVECs were treated with 50 ng mL$^{-1}$ hIFN-γ for 72 h to stimulate expression of MHC class II prior to the addition of T cells. In all, $1.5 \times 10^6$ CFSE labeled CD4 + memory T cells were added to $1 \times 10^5$ endothelial

cells and cultured RPMI media supplemented with 10% FBS, 2 mM L-glutamine, 100 U mL$^{-1}$ penicillin, 100 ug mL$^{-1}$ streptomycin for 7 days. After co-culture, T cells were collected, washed in PBS, and stained for human CD4 + (clone RRA-T4, Biolegend) in FACs-staining buffer (2% BSA, PBS) for 30 min on ice. All samples were analyzed on the Attune NxT Flow Cytometer.

**Transplantation of human arterial allografts.** Animal experiments were performed in accordance with guidelines of Yale Institutional Animal Care and Use Committee. Human arteries were obtained under a protocol approved by the Yale Human Investigations Committee. 1–3 mm segments of human epicardial coronary arteries were transplanted into the infra-renal aorta of female C.B-17 SCID/beige mice (Taconic Biosciences) by end-to-end microsurgical anastomotic technique and healed-in for 30 days, as described previously[10, 29, 42]. Adjacent artery segments were transplanted into 2–5 mice for each experiment, and data from individual experiments were pooled for analysis. Healed-in human arteries, along with 1–2 mm cuff of mouse aorta on each side, were harvested and incubated in M199 culture medium containing 0.2 mg mL$^{-1}$ CIITA siRNA nanoparticles (60 nM) or CTL siRNA nanoparticles for 6 h at 37° C. After 6 h, arterial grafts were washed with DPBS × 3 and re-transplanted into the infra-renal aorta of a second SCID/bg mouse. 1 day after transplantation, recipients were injected with 200 ng of human IFN-γ (IP) every other day for 1 week to induce expression of MHC class II. For 2, 4, and 6 week experiments, the first dose of hIFN-γ was injected 6 days before harvest. Two days after the final hIFN-γ injection, arterial grafts were harvested.

**Transplantation of human arterial allografts into PBMC reconstituted mice.** In all, 1–3 mm segment of human epicardial coronary artery was transplanted into the infrarenal aorta of immunodeficient C.B-17 SCID/bg mouse and healed-in for 30 days. Prior to re-transplantation, $1.5 \times 10^8$ human PBMCs (allogeneic to the arterial graft) were injected into another SCID/bg mouse and the level of circulating human CD3 + T cell was assessed 10 days after PBMC administration. To measure T-cell engraftment, blood was collected via retro-orbital puncture and stained with FITC conjugated anti-human CD3 antibody and 647 conjugated-anti-mouse CD45 antibody. 0.5% to 10% of human T cells of the total mononuclear cell population was considered as successful engraftment. At the time of re-transplantation, arteries were removed, incubated in M199 media containing siRNA-loaded PACE nanoparticles for 6 h and transplanted into SCID/bg mice reconstituted with human PBMCs. Arteries were harvested 10 days after re-transplant. n = 4 for each experimental group.

**Immunofluorescence.** Human arterial grafts were stained with mouse anti-human HLA-DR antibody (Abcam), rabbit anti-human CD31 antibody (Abcam), FITC-conjugated α-SMA (Sigma), mouse anti-human CD45RO (eBioscience), Alexa Fluor 488-conjugated goat anti-mouse secondary antibody (Invitrogen), and Alexa Fluor 555-conjugated goat anti-rabbit secondary antibody (Invitrogen). Tissue sections were fixed in 4% paraformaldehyde (Sigma) for 15 min, blocked using 10% donkey serum in PBS, incubated overnight at 4 °C in primary antibodies, and incubated for 1 h at room temperature in secondary antibodies. Sections were imaged on the confocal microscope. To quantify HLA-DR (MHC class II) knockdown, hCD31 positive cells were isolated using image J. In this cell population, total intensity of the HLA-DR signal was quantified using Image J and normalized to CTL nanoparticle treated arterial grafts.

**Statistics.** All studies were conducted in triplicates and repeated independently. Result are expressed as mean ± standard deviation (SD). Statistical analysis was performed with Prism software (GraphPad) using Student's unpaired $t$-test and one-way ANOVA. $P$-values < 0.05 was considered statistically significant.

**Study approval.** All human cells, umbilical cords, and arteries were obtained under protocols approved by Yale Human Investigation Committee and the New England Organ Bank. All animal protocols were approved by the Yale University Institutional Animal Care and Use Committee.

**Data availability.** The data that support the findings of this study are available within the article and its supplementary information files and from the corresponding author upon reasonable request.

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

## Acknowledgements

We thank Louise Benson and Gwendolyn Davis-Arrington for assistance with cell culture and animal care. This work was supported by grants from the National Institutes of Health (AI126166, AI106992 and AI112443). J.C. was supported by NIH National Research Service Award predoctoral fellowship (F31HL132469).

## Author contributions

J.C., J.S.B., and W.M.S, conceived and designed experiments. J.Z. synthesized polymers used in this paper. J.C. performed nanoparticle preparations. J.C. and P.A. performed in vitro studies. J.C. and H.L. performed ex vivo studies. J.C., L.Q., G.L., and G.T. designed and conducted animal studies. J.C. and G.T.T. analyzed data. J.C., J.S.B., and W.M.S. co-wrote the paper. All authors edited the paper.

## Additional information

**Competing interests:** The authors declare no competing financial interests.

