## [Peer Review File · Nature Communications]

Reviewers' comments:

Reviewer #1 (Remarks to the Author):

Cui et al. report on a novel nanotechnology based technique to silence MHC class II expression on human umbilical vein endothelial cells ex vivo. The authors optimized a CIITA siRNA-loaded PACE nanoparticles to perfuse human umbilical veins and decrease their MHC class II expression. Those veins were later transplanted into immunodeficient mouse hosts to show that they are less immunogenic by reducing alloimmune CD4 T cell proliferation in vivo.

This is an interesting study by a group who recently published similar work in JCI insight (Abrahimi et al, January 21, 2016). They also reduced MHC class II expression on human artery grafts using CRISP/Cas9 technology or siRNA knockdown, that resulted in less immunogenicity and lower CD4 and CD8 T cell activation. The difference in this study is the novel nanotechnology approach to silence the MHC class II.

Major concerns:

1- My major concern is the biologic significance of reducing Class II MHC molecules for 2 weeks on solid organ transplant outcomes as far as acute and chronic rejection.

a- The concept of reducing MHC class II to reduce immunogenicity of allografts is not novel. Multiple papers tried to answer this question since the 1990s. However, some found that MHC class II expression is required to achieve long term survival in heart transplant vascularized model after Co-stimulatory blockade. This shows a role for MHC class II in immune-regulation (Yamada et al 2001, Journal of Immunology) that is the most important for long-term outcome. This should be discussed by the authors and those papers should be referenced.

b- Silencing MHC class II on endothelial cells of arteries or veins may reduce allo-immunity and may be important in arteries or veins allo-transplantation but the authors do not have the data that this approach will make any biologic difference in acute or chronic rejection of solid organ transplant. Donor dendritic cells from solid organ transplants are found in secondary lymphoid tissues within 3 hours of transplantation and they play critical role in T cell priming. I would have been much more excited about this paper if the authors used a vascularized heart transplant model in mice and showed an effect of silencing Class II MHC molecules on acute or chronic rejection. To test the effect on acute rejection: They can perfuse BALB/c hearts ex vivo and transplant them into C57BL/6 mice. For chronic rejection: They can perfuse BM12 hearts and transplant them into C57BL/6 mice.

c- The readout for the allo-immunity is very weak to make any conclusions. The authors can use the same model they used in their JCI paper to show reduction in T cell mediated destruction of the vessels in vivo and do more thorough characterization of the alloimmune response to the transplanted veins.

d- While reducing the MHC class II expression on endothelial cells for 2 weeks is important for acute rejection, the major hurdle for improving outcomes in transplanted patients is chronic rejection as the rate of acute rejection is minimal with the current immunosuppressive modalities. Again no data to show how biologically significant it is to silence those molecules for 2 weeks on chronic rejection that occurs months to years after transplant in humans (weeks to months after transplant in mice). It is not enough to say that acute rejection puts patients at risk for chronic rejection.

2- The authors show short-term toxicity data. However, the siRNA is present in the tissues for more than 10 days. What is the toxicity at 10 days ex vivo. It will be helpful to show the long-term integrity of the perfused vessels in vivo (beyond 2 weeks).

3- The authors show no toxicity for PACE-90. However, for the ex vivo perfusion experiments of the umbilical vein, they used the PACE-70 that has better silencing efficiency but higher toxicity to the endothelial cells than PACE-90 (30% vs. 5 % respectively). How much this toxicity affects the integrity of the vessel and the ability of endothelial cells to stimulate T cells independent of the expression of Class II MHC molecules. No in vivo data to show the integrity of the vessels few weeks after transplant in the presence of allo-immunity.

Minor comments:

1- Endothelial cells express cell adhesion molecules (selectins and integrins) to facilitate transmigration of the innate and adaptive immunity to the allograft. It will be interesting to know how the stress of siRNA transduction affects the expression of Selectins and Integrins on endothelial cells.

2- It will be interesting to show representative figures of the experiments in Figures 2C and 2D.

3- The authors should give more details of the experiments done in the result section so the readers don't have to go back and forth to understand the methodology used. This applies in particular to figures 4 and 5.

4- Figure 6 legend: anti-CD3 should be anti-CD31

Reviewer #2 (Remarks to the Author):

This is a brief technical report showing that ex vivo using CIITA siRNA-loaded PACE nanoparticles dramatically attenuate MHC class II expression on endothelial cells.

For the most part, the data are clear and of high quality, and the manuscript is well organized and well written.

The major difficulty with the data is that the effects of PACE-70 seemed to last up to 10days, a time at which most allografts are completely rejected or nearly so. If so, how do you extend its effect to have graft prolongation with using the compound? The clinically relevant point is, can this technique be used to reduce T cell activation via less antigen presentation regulated by the management of endothelial cells and provide graft prolongation with using co-stimulatory blockades such as PD-L1 and PD-L2? During ischemic time (ex vivo), how about CD80/CD86 expressions on the endothelial cells? Longer TIT, less graft survival. Again, the clinically relevant point is that rejections in humans occur in the setting of on-going immunosuppression and a modified immune response. The current method with using the compound would provide the same result following 24-48hours (4dgree) reservation?

Minor point: any chance to perform the method in chronic model?

Reviewer #3 (Remarks to the Author):

Endothelial cell expression of donor MHC class II molecules has been shown to be able to directly activate alloreactive memory CD4 T cells which provide "help" for CD8 memory T cells to develop into CTL that damage the endothelium. Previous work from the authors have shown that if MHC class II is blocked or it's expression is diminished via CRISPR/Cas9 on human vessels, transplanted to immunodeficient mice reconstituted with human immune cells, rejection is diminished. The current manuscript examines the capacity of CIITA targeting siRNA to be delivered to donor endothelium by poly(amine-co-ester) nanoparticles and it's subsequent impact on MHC class II expression using immunodeficient mice.

This is a well-written manuscript that is clear and easy to follow. To my mind, the main novelty here is the delivery system and the potential to modify expression in donor endothelial cells during

normothermic machine perfusion. With this in mind the authors could have performed all of these studies looking at any molecule expressed by endothelial cells as no functional studies were performed to look at the impact of loss of MHC class II from vessel endothelium. This data would have strengthened the manuscript and as knockdown is not complete vessels may still express a small amount of MHC class II that is enough to activate T cells in vivo (although I accept that the response is poor in vitro).

In a way using knockdown of donor MHC class II has led the authors to broach the subject of whether this is clinically important; although one of the authors papers has been cited showing a role for donor MHC class II on graft endothelium in rejection there are a number of other papers that suggest it is unlikely to be one of the main drivers of rejection (with donor DC and donor-derived exosomes being more important and presumably being untouched by the nanoparticles). Ultimately, I found that using knockdown of CIITA the MHC class II (and discussion) a little distracting from the main message which is that the nanoparticles could be used to alter the expression of several immunodulatory molecules on donor endothelium which may have a marked impact on effector and memory T cell infiltration of transplants.

Specific points

(1) Figure 4– How does the ml/minute that allowed successful nanoparticle take up compare to the flow rates that are currently being used in organs receiving normothermic machine perfusion? This is important as if the nanoparticles are not taken up under conditions necessary to sustain/repair organ transplants then this would be a major barrier to using this sort of delivery system in such a way.

(2) Most figures give no indication of how many times the experiment was repeated or if an experiment was repeated more than once. Most data is presented without a statistical analysis. This information should be placed in the legends.

(3) In figure 6 – Why did authors only look at 14 days post transplantation? It would seem if you are getting sustained siRNA-mediated knock-down it would be even more advantageous and impressive to look at later time-points and see whether this is prolonged even further.

***Ex vivo* pretreatment of human vessels with siRNA-loaded nanoparticles provides long-lasting protein silencing in endothelial cells**

Reviewer #1

1- My major concern is the biologic significance of reducing Class II MHC molecules for 2 weeks on solid organ transplant outcomes as far as acute and chronic rejection.

We agree with the point that elimination of class II MHC expression on graft endothelial cells will not, on its own, eliminate rejection and we have made this point more explicit in the revised manuscript. However, as pointed out by reviewer 3, the central point we wish to make in this report is that *ex vivo* PACE nanoparticle-mediated siRNA transfection is safe and can provide sustained effects in graft endothelial cells, which are still measurable at 6 weeks. We have rewritten the Introduction to make this point more clearly.

a- The concept of reducing MHC class II to reduce immunogenicity of allografts is not novel. Multiple papers tried to answer this question since the 1990s. However, some found that MHC class II expression is required to achieve long term survival in heart transplant vascularized model after Co-stimulatory blockade. This shows a role for MHC class II in immune-regulation (Yamada et al 2001, Journal of Immunology) that is the most important for long-term outcome. This should be discussed by the authors and those papers should be referenced.

We thank the reviewer for citing this reference, Yamada and colleagues demonstrated that elimination of the direct pathway (knockdown of MHC class II in donors) results in prolonged allograft survival whereas elimination of the indirect pathway (knockdown of MHC class II in recipients) made it more difficult to achieve prolonged allograft survival. It is critical to note that the class II-expressing graft cells involved in direct presentation are passenger leukocytes, which are not targeted by intravascular PACE transfection. More importantly, mouse models such as those used by Yamada et al., differ in critically important ways from humans in that mouse endothelial cells are not able to activate CD4+ effector T cell responses. Furthermore, as we noted above, we are not arguing that this single manipulation can prevent rejection, but rather that manipulating graft ECs can be achieved by PACE nanoparticle-mediated transfection and we have elected to make this point using a target siRNA we had previously validated *in vitro*.

b- Silencing MHC class II on endothelial cells of arteries or veins may reduce allo-immunity and may be important in arteries or veins allo-transplantation but the authors do not have the data that this approach will make any biologic difference in acute or chronic rejection of solid organ transplant. Donor dendritic cells from solid organ transplants are found in secondary lymphoid tissues within 3 hours of transplantation and they play critical role in T cell priming. I would have been much more excited about this paper if the authors used a vascularized heart transplant model in mice and showed an effect of silencing Class II MHC molecules on acute or chronic rejection. To test the effect on acute rejection: They can perfuse BALB/c hearts *ex vivo* and transplant them into C57BL/6 mice. For chronic rejection: They can perfuse BM12 hearts and transplant them into C57BL/6 mice.

We appreciate the reviewer's perspective. We agree that transplantation of a solid organ (i.e. vascularized heart transplant) to study acute and chronic rejection will be important to understand clinical value of our approach. However, as we have noted above and emphasized in

the revised manuscript, mouse models simply cannot be used to evaluate the contribution of human graft endothelium to rejection. It is also important to note that passenger leukocytes are probably eliminated very soon after transplantation whereas human, unlike mouse grafts, retain class II MHC expressing endothelial cells. This may explain why indirect presentation becomes so important in mouse models.

c- The readout for the allo-immunity is very weak to make any conclusions. The authors can use the same model they used in their JCI paper to show reduction in T cell mediated destruction of the vessels *in vivo* and do more thorough characterization of the alloimmune response to the transplanted veins.

This is a fair point. We have added a new experiment that shows suppression of MHC class II on arteries results in a reduction in intimal expansion, endothelial cell destruction, and T cell infiltration after 10 days.

d- While reducing the MHC class II expression on endothelial cells for 2 weeks is important for acute rejection, the major hurdle for improving outcomes in transplanted patients is chronic rejection as the rate of acute rejection is minimal with the current immunosuppressive modalities. Again no data to show how biologically significant it is to silence those molecules for 2 weeks on chronic rejection that occurs months to years after transplant in humans (weeks to months after transplant in mice). It is not enough to say that acute rejection puts patients at risk for chronic rejection.

The reviewer makes an important point that late graft loss is not well controlled by current therapies and we are focusing on perioperative events. However, it is well established in both patients and animal models that early events have an impact on long term graft survival, most likely by shifting the balance between effector and regulatory arms of the adaptive immune response. We do not know if reduction of class II MHC molecules on human graft endothelial cells is going to be useful, a conclusion that goes well beyond the current study. However, we believe that we have convincingly demonstrated the value of PACE nanoparticles in this setting and these could be used for many different targets.

2- The authors show short-term toxicity data. However, the siRNA is present in the tissues for more than 10 days. What is the toxicity at 10 days *ex vivo*. It will be helpful to show the long-term integrity of the perfused vessels *in vivo* (beyond 2 weeks).

We did not see toxicity *in vitro* at 10 days and did not carry out longer term culture experiments because the nanoparticles were simply diluted out by cell division during cell culture. To address this point, we have added new experiments showing that transduced graft endothelial cells remain viable *in vivo* after 6 weeks.

3- The authors show no toxicity for PACE-90. However, for the *ex vivo* perfusion experiments of the umbilical vein, they used the PACE-70 that has better silencing efficiency but higher toxicity to the endothelial cells than PACE-90 (30% vs. 5 % respectively). How much this toxicity affects the integrity of the vessel and the ability of endothelial cells to stimulate T cells independent of the expression of Class II MHC molecules. No *in vivo* data to show the integrity of the vessels few weeks after transplant in the presence of allo-immunity.

As stated above, we have added new experiments that show long term viability of endothelial cells in the transfected grafts.

Minor comments:

1- Endothelial cells express cell adhesion molecules (selectins and integrins) to facilitate transmigration of the innate and adaptive immunity to the allograft. It will be interesting to know how the stress of siRNA transduction affects the expression of Selectins and Integrins on endothelial cells.

We thank the reviewer for this suggestion. We assume the reviewer meant integrin ligands, as these are the endothelial molecules involved in leukocyte recruitment. We examined the expression of E-selectin and integrin ligand VCAM-1 (proteins that participate in leukocyte recruitment) in cultured ECs after NP pretreatment. NP pretreatment did not affect EC expression of E-selectin or VCAM-1 compared to PBS controls (see below). This data has been added to the Supplementary figures (see Figure S1).

2- It will be interesting to show representative figures of the experiments in Figures 2C and 2D.

We thank the reviewer for this suggestion. We have included representative plots for PACE-70 nanoparticles from Figures 2C and 2D (see below). We have added these figures to the Supplemental Data (see Figure S3).

3- The authors should give more details of the experiments done in the result section so the readers don't have to go back and forth to understand the methodology used. This applies in particular to figures 4 and 5.

As suggested, we have revised the Results section to include more details on methods used in these studies.

4- Figure 6 legend: anti-CD3 should be anti-CD31

We thank the reviewer for pointing out this oversight. We have modified the caption for Figure 6.

Reviewer #2

The major difficulty with the data is that the effects of PACE-70 seemed to last up to 10 days, a time at which most allografts are completely rejected or nearly so. If so, how do you extend its effect to have graft prolongation with using the compound?

As we noted in the manuscript, 10 days is the limit that we can test during in vitro experiments because the nanoparticles are diluted out due to cell division. (Inhibition of cell division for 10 days in culture is toxic for the cells.) However, as noted in the text, endothelial cells typically divide very slowly in vivo, allowing a more sustained time course. Our new experiments show that siRNA effects are still evident at 6 weeks post *ex vivo* transfection.

The clinically relevant point is, can this technique be used to reduce T cell activation via less antigen presentation regulated by the management of endothelial cells and provide graft prolongation with using co-stimulatory blockades such as PD-L1 and PD-L2?

We have added new experiments to show that suppression of MHC class II on arteries resulted in a reduction in intimal expansion, endothelial cell destruction, and T cell infiltration after 10 days. This method could be applied to regulate other immunomodulatory molecules on donor endothelial cells, which could further improve graft survival after transplantation.

During ischemic time (*ex vivo*), how about CD80/CD86 expressions on the endothelial cells?

In our hands, human endothelial cells do not express CD80 or CD86

The current method with using the compound would provide the same result following 24-48hours (4degree) reservation?

We have previously shown that nanoparticle uptake involves clathrin-mediated endocytosis and would be unlikely to work in the cold. However, as noted in the Discussion, transfection is compatible with ex vivo normothermic machine perfusion, an approach we are actively pursuing.

Minor point: any chance to perform the method in chronic model?

For reasons cited in response to reviewer 1, we are focused on targeting human and not mouse endothelium. We are unaware of any simple model of chronic human rejection.

Reviewer #3:

With this in mind the authors could have performed all of these studies looking at any molecule expressed by endothelial cells as no functional studies were performed to look at the impact of loss of MHC class II from vessel endothelium. This data would have strengthened the manuscript and as knockdown is not complete vessels may still express a small amount of MHC class II that is enough to activate T cells in vivo (although I accept that the response is poor in vitro).

In this revised manuscript, we have added new experiments to examine allograft rejection in the presence of human T cells. Suppression of MHC class II on arteries resulted in a reduction in intimal expansion, endothelial cell destruction, and T cell infiltration after 10 days.

In a way using knockdown of donor MHC class II has led the authors to broach the subject of whether this is clinically important; although one of the authors papers has been cited showing a role for donor MHC class II on graft endothelium in rejection there are a number of other papers that suggest it is unlikely to be one of the main drivers of rejection (with donor DC and donor-derived exosomes being more important and presumably being untouched by the nanoparticles). Ultimately, I found that using knockdown of CIITA the MHC class II (and discussion) a little distracting from the main message which is that the nanoparticles could be used to alter the expression of several immunodulatory molecules on donor endothelium which may have a marked impact on effector and memory T cell infiltration of transplants.

We agree that the main message should be proof of concept, and we have revised the manuscript to reflect this emphasis. Regarding the reviewer's other point, however, class II MHC on endothelium is likely to be much more important than can be appreciated in mouse models. Human EC can be a source of exosomes or nanoparticles that continue to activate the direct pathway, something that would not be expected in mouse models.

Specific points

(1) Figure 4– How does the ml/minute that allowed successful nanoparticle take up compare to the flow rates that are currently being used in organs receiving normothermic machine perfusion? This is important as if the nanoparticles are not taken up under conditions necessary

to sustain/repair organ transplants then this would be a major barrier to using this sort of delivery system in such a way.

Nicholson and colleagues have perfused kidneys at a mean renal blood flow rate of 67 ± 29 mL/min/100g. This flow rate was titrated up over a period of several hours. If we were to compare this flow rate to our *ex vivo* perfusion of the umbilical vein, we can approximate the optimal flow rate for maximal particle uptake to be ~ 100 mL/min/100g. It is also important to note the flow rate in each blood vessel in an organ (i.e. kidneys) is not uniform and varies as a function of diameter.

(2) Most figures give no indication of how many times the experiment was repeated or if an experiment was repeated more than once. Most data is presented without a statistical analysis. This information should be placed in the legends.

We thank the reviewer for noticing this omission, and we have modified the legends to include statistical information.

(3) In figure 6 – Why did authors only look at 14 days post transplantation? It would seem if you are getting sustained siRNA-mediated knock-down it would be even more advantageous and impressive to look at later time-points and see whether this is prolonged even further.

We have performed new studies to look at 4 weeks and 6 weeks post-transplant. This data is now included in the manuscript.

REVIEWERS' COMMENTS:

Reviewer #1 (Remarks to the Author):

No further comments.

Reviewer #2 (Remarks to the Author):

The revised manuscript is another nice example of the integration of immunobiology and biotechnology techniques.

The manuscript is well written, excellently organized and deals with an important and timely topic.

Reviewer #3 (Remarks to the Author):

No further comments